# Transcriptome Profiling Provides Insights into the Early Development of Tiller Buds in High- and Low-Tillering Orchardgrass Genotypes

**DOI:** 10.3390/ijms242216370

**Published:** 2023-11-15

**Authors:** Guangyan Feng, Xiaoheng Xu, Wen Liu, Feigxiang Hao, Zhongfu Yang, Gang Nie, Linkai Huang, Yan Peng, Shaun Bushman, Wei He, Xinquan Zhang

**Affiliations:** 1College of Grassland Science and Technology, Sichuan Agricultural University, Chengdu 611130, China; 2Forage and Range Research Laboratory, United States Department of Agriculture, 695 North 1100 East, Logan, UT 84322-6300, USA; 3Grassland Research Institute, Chongqing Academy of Animal Science, Chongqing 402460, China

**Keywords:** perennial forage, orchardgrass, tillering regulation, photosynthesis, transcriptome profiling

## Abstract

Orchardgrass (*Dactylis glomerata* L.) is among the most economically important perennial cool-season grasses, and is considered an excellent hay, pasture, and silage crop in temperate regions worldwide. Tillering is a vital feature that dominates orchardgrass regeneration and biomass yield. However, transcriptional dynamics underlying early-stage bud development in high- and low-tillering orchardgrass genotypes are unclear. Thus, this study assessed the photosynthetic parameters, the partially essential intermediate biomolecular substances, and the transcriptome to elaborate the early-stage profiles of tiller development. Photosynthetic efficiency and morphological development significantly differed between high- (AKZ-NRGR667) and low-tillering genotypes (D20170203) at the early stage after tiller formation. The 206.41 Gb of high-quality reads revealed stage-specific differentially expressed genes (DEGs), demonstrating that signal transduction and energy-related metabolism pathways, especially photosynthetic-related processes, influence tiller induction and development. Moreover, weighted correlation network analysis (WGCNA) and functional enrichment identified distinctively co-expressed gene clusters and four main regulatory pathways, including chlorophyll, lutein, nitrogen, and gibberellic acid (GA) metabolism pathways. Therefore, photosynthesis, carbohydrate synthesis, nitrogen efficient utilization, and phytohormone signaling pathways are closely and intrinsically linked at the transcriptional level. These findings enhance our understanding of tillering in orchardgrass and perennial grasses, providing a new breeding strategy for improving forage biomass yield.

## 1. Introduction

Orchardgrass (*Dactylis glomerata* L.) is among the top four economically important cool-season forage grasses worldwide due to its high biomass and broad adaptability [1]. In southwest China, orchardgrass is grown primarily for pasture, but this grass is also used for hay and silage in other regions. The orchardgrass tiller is a basal branch on the thickened lower crown near the ground. Tillering performance is an important agronomic trait and a determinant of rational close planting, which directly affects forage biomass. Moreover, tillering is associated with the growth habit of perennial grasses, especially by regulating forage harvesting and rangeland grazing. The most important role of tillers is regenerating perennial forage grasses for conservation after the summer or winter dormancy. The number and size of remaining vegetative tillers after cutting or livestock feeding is related to the level of plant regrowth.

In contrast, specific developmental processes, such as inflorescence formation and flowering, may inhibit tillering, even under ideal environments. These processes mainly depend on the accumulation and redistribution of photosynthetic products, especially natural macromolecules such as carbohydrates [2]. Maize and rice have more carbohydrates for tillering during the early vegetative phase, while low carbohydrate content may inhibit tillering in wheat [3,4,5]. Under natural conditions, efficient nitrogen (N) uptake increases cytokinin concentrations in the tiller nodes, further promoting tiller primordium germination and tiller formation [6]. The differences in the tillering potentials of grasses may be attributed to the temperature, light, fertilization, and carbon status of grasses [7]. Consequently, this study aimed to reveal the transcriptional difference between high- and low-tillering orchardgrass genotypes.

Studies have unraveled the molecular basis of branching and tillering in sorghum (*Sorghum bicolor*), pea (*Pisum sativum*), Arabidopsis (*Arabidopsis thaliana*), and rice (*Oryza sativa*) [8,9,10,11,12]. The tillering mechanisms of perennials are more complex and unpredictable due to the life cycle difference between annuals and perennials. For instance, in a perennial crop such as orchardgrass, tillers originate annually from the crown at the beginning of the growing period in the yearly life cycle. The crop then undergoes a vegetative growth phase before the reproductive phase of flowering and seed maturation, and eventually enters senescence and dormancy in summer. Nevertheless, cereals and grass tillering programs still have similarities, offering a potential framework to understand these processes in orchardgrass.

In rice, modeling tiller or branch formation involves several specific regulators, and this process also implicates genes related to phytohormone and lignin biosynthesis. Meristems’ initiation/maintenance was severely constrained in rice the LAX PANICLE1 (*lax1*) mutant, suggesting that the *LAX1* gene is required for initiating/maintaining axillary meristems [13]. Recent studies revealed that the *REGULATOR OF AXILLARY MERISTEM FORMATION* (*ROX*) in *Arabidopsis* is the ortholog of rice LAX1, a key regulator of axillary meristem initiation [14]. The *MONOCULM 1* (*MOC1*) contains a GRAS (GIBBERELLIC-ACID INSENSITIVE, REPRESSOR of GAI and SCARECROW) domain, the first functionally defined gene that controls tillering in rice. The *moc1* mutant had reduced tillering ability, producing only one main culm, thus showing that MOC1 positively regulates rice tillers [15]. The rice *homeobox 1* (*OSH1*) and *TEOSINTE BRANCHED1* (*TB1*) are direct downstream *MOC1* genes [16,17], and *TAD1*/*TE* regulates tillering by degrading the MOC1 protein [18,19]. The *lax1 moc1* double mutant exhibited more panicle branch and tiller deficiency, indicating the potential function overlap of LAX1 and MOC1 in tillering [20].

Furthermore, genes also regulate plant tillering through phytohormone signaling. Tillering is inhibited by gibberellic acid (GA), and the overexpressing of GA2oxs can induce a high tiller number [21,22]. GA has recently been reported to trigger the degradation of SLENDER RICE 1 (SLR1), resulting in plants with reduced tillers [23]. In addition, *DWARF3* (*D3*), *DWARF14* (*D14*), and *Ideal Plant Architecture 1* (*IPA1*) may play a critical role in repressing tillering by disrupting strigolactone (SL) perception and signaling, and *D53* represses SLs signaling through the D14-SCF-D3 ubiquitination complex. Contrarily, *DWARF10* (*D10*), *DWARF27* (*D27*), and *DWARF17* (*D17*) regulate tillering regulation through SL biosynthesis [24,25,26,27]. Research indicated that *DHTA-34* is allelic to *D14*, and mutant *dhta-34* showed reduced sensitivity to SLs and increased tillering number [28]. In petunia, the tillering-associated gene *DAD2* also regulates SLs [29]. Cytokinin is a key regulator of lateral bud activation, whereas IAA inhibits lateral bud release [30,31]. Furthermore, the brassinosteroid-associated gene *DWARF AND LOW TILLERING* (*DLT*), *BR-INSENSITIVE 1* (*HvBRI1*), and miR156 mediated-*TaSPLs* are potentially associated with tillering in cereals [32,33,34]. Although research has provided clues into the tillering regulatory network, the molecular mechanism of tillering in perennials remains largely unknown.

The assembly and release of the orchardgrass reference genome provided reliable, functional information and regulatory networks based on high-throughput transcriptomic data, which serve as a considerable and efficient avenue for mechanistic investigations. A gene set-oriented approach integrates direct experimental evidence and RNA-seq data for exploring upstream regulatory factors. This approach was utilized in previous studies on flowering time regulation, tiller formation, and heat/drought stress response in orchardgrass [1,35,36,37,38,39,40]. A high-throughput RNA-seq of multiple tissues at the tillering stage of high- and low-tillering orchardgrass genotypes revealed that the discriminatory efficiencies of SLs, ABA, and GA biosynthesis greatly influence orchardgrass tillering, especially the SLs [36]. Moreover, a systematic analysis of the tillering-associated GRAS gene family also provided insights into their potential functions in orchardgrass growth and development [41]. Although these studies provide clues to tillering regulation in orchardgrass, their main conclusions are based on the tillering mechanisms of annual model crops. This study generated a continuous transcriptome, elaborated the gene expression changes at the early stages of tillering bud development, constructed the co-expression network of genes responding to tillering regulation, and uncovered important regulatory pathways affecting the orchardgrass tillering potential. The results facilitate the understanding of the tillering mechanisms of perennial grasses and selective breeding strategies for high-yield perennial forage grasses.

## 2. Results

### 2.1. Photosynthetic Parameters and Chlorophyll Content

The total Chl content and Chl a/b ratio significantly differed between the high- and low-tillering orchardgrass. High-tillering orchardgrass had higher total Chl contents and lower Chl a/b ratio than low-tillering orchardgrass (Figure 1g,h). A previous report proposed that limited nitrogen in leaves increased the Chl a/b ratio and decreased the Chl content in leaves [42]. High-tillering orchardgrass had significantly higher stomatal conductance (Gs), net photosynthetic rate (Pn), transpiration rate (Tr), Fv/Fm, and PIABS than low-tillering orchardgrass (Figure 1a–c,e,f). However, the WUE was not significantly different between the high- and low-tillering orchardgrass (Figure 1d).

### 2.2. Lutein, Zeaxanthin, Glutamate, Glutamine, and GA Concentration

Low-tillering orchardgrass had significantly higher lutein and zeaxanthin concentrations (Figure 2a,b), and accumulated more glutamate and glutamine in their leaves (Figure 2c,d) than high-tillering orchardgrass. Lutein and zeaxanthin have similar oxidant photoprotection and detoxification roles in photosynthetic organisms. High-tillering orchardgrass accumulated significant or extremely significant levels of GA3, GA4, and GA5 than their low-tillering counterparts (Figure 2f–h); however, low-tillering orchardgrass had higher GA1 and GA6 levels than the high-tillering (Figure 2e,i). High-tillering orchardgrass had higher GA7 levels than the low-tillering ones, but the difference was insignificant (Figure 2j).

### 2.3. RNA-Seq Data Generation

Thirty samples (three biological replicates at five sampling points of two genotypes) were collected for consecutive and comprehensive transcriptome analysis of high-(AKZ-NRGR667) and low-tillering orchardgrass (D20170203). The purpose was to obtain an overview of the *D. glomerata* transcriptome data at different developmental stages. A total of 206.41 Gb of data were generated, with an average of 6.88 Gb per sample (Appendix A). The raw sequence data have been deposited in the Genome Sequence Archive (Genomics, Proteomics & Bioinformatics 2021) at the National Genomics Data Center (Nucleic Acids Resarch 2022), China National Center for Bioinformation/Beijing Institute of Genomics, Chinese Academy of Sciences (GSA: CRA011917) and are publicly accessible (https://ngdc.cncb.ac.cn/gsa, accessed on 20 July 2023). After trimming, each sample had 43.5 M–52.1 M high-quality clean reads whose Q30 values exceeded 93% (Appendix A). Over 88.24% of the clean reads were mapped to the reference genome, of which 84.98% were mapped to unique genomic locations (Appendix A). Moreover, 87.86, 3.89, and 8.25% of the clean reads mapped to exons, introns, and intergenic regions, respectively (Appendix A). Figure 3a,b show the gene expression and fragments per kilobase of transcript sequence per million (FPKM) distribution of the 30 samples. The 30 samples clustered in 10 corresponding discrete groups, and group A_L4 showed a longer distance than other groups (Figure 3c,d). The sequencing results confirmed the high quality of the generated sequence data, which was suitable for subsequent analyses of biological characteristics.

### 2.4. Identification and Functional Annotation of Differentially Expressed Genes (DEGs)

A total of 17,415 DEGs were detected in four pairwise developmental-stage comparisons of high-tillering orchardgrass genotypes. A_L4 vs. A_L3 (A_P4) had remarkably more DEGs (5710 up-regulated and 4029 down-regulated) than other pairwise comparisons of the high- and low-tillering materials, suggesting that L4 and L3 phases of AKZ-NRGR667 have complex transcriptional processes. Moreover, A_L1 vs. A_HB had 4070 DEGs (2289 up-regulated and, 1781 down-regulated), while A_L2 vs. A_L1 had 1004 (702 up-regulated and, 302 down-regulated), and A_L3 vs. A_L2 had 2602 (1251 up-regulated and, 1351 down-regulated) (Figure 4). The low-tillering orchardgrass (P3) exhibited the maximum number of DEGs (5671) earlier than the high tillering orchardgrass (P4) (9739) (Figure 4). At P3, D20170203 had approximately twice the number of DEGs in AKZ-NRGR667. However, AKZ-NRGR667 had more DEGs at P4 than D20170203. Additionally, all comparisons had more up-regulated genes than down-regulated ones, except A_L3 vs. A_L2. The five examined AKZ-NRGR667 developmental stages had more DEGs, with the highest being in L4 compared to the L3 and L4 of D20170203. These results show that the transcriptional difference in time scales may cause different tillering abilities between the orchardgrass varieties.

The high- and low-tillering orchardgrass DEGs enriched diverse functional GO and KEGG terms in the four developmental phases. The DEGs at A_P1 significantly enriched photosynthesis or photosynthesis-related categories (Appendix A). Similarly, the same DEGs at A_P1 significantly enriched pathways of photosynthesis-antenna proteins, photosynthesis, carotenoid biosynthesis, porphyrin and chlorophyll metabolism, carbon fixation in photosynthetic organisms, and nitrogen metabolism (Appendix A). The DEGs in D20170203 significantly enriched different functional categories involving oxidoreductase activity, especially sucrose metabolic processes such as sucrose synthase activity and UDP-glycosyltransferase activity (Appendix A). Nevertheless, the DEGs in D20170203 significantly enriched nitrogen metabolism pathways, cutin, suberine, wax biosynthesis, and phenylpropanoid biosynthesis at P1 (Appendix A).

In the second phase (L2 vs. L1), the DEGs significantly enriched GO terms for the amino sugar metabolic process, chitin metabolic process, and cell wall macromolecule metabolic process of AKZ-NRGR667 (Appendix A). However, more L2 vs. L1 DEGs (of D20170203) enriched GO terms related to stress response, including response to inorganic substance, abiotic stimulus, oxygen-containing compound, acid chemical, and water (Appendix A). The DEGs also enriched the signal transduction GO term in the second phase. The MAPK signaling pathway was significantly enriched in AKZ-NRGR667, while circadian rhythm and phenylpropanoid synthesis were significantly enriched in D20170203 (Appendix A).

In the third phase (L3 vs. L2), the number of DEGs in D20170203 far exceeded those in AKZ-NRGR667. The DEGs significantly enriched GO and KEGG photosynthesis or photosynthesis-related terms and pathways. In addition, GO terms for stress response, such as response to auxin, response to abiotic stimulus, and response to oxidative stress, were significantly enriched (Appendix A). The DEGs significantly enriched the plant hormone signal transduction, phenylpropanoid biosynthesis, carotenoids biosynthesis, nitrogen metabolism, and brassinosteroids biosynthesis pathways in phase D_P3 (Appendix A). In contrast, P4 of AKZ-NRGR667 had the most numerous DEGs that significantly enriched pathways related to photosynthesis, stress response, and phenylpropanoid biosynthesis (Appendix A).

In summary, the regulatory processes enriched by the DEGs were temporally inconsistent between the high- and low-tillering orchardgrass. The DEGs in AKZ-NRGR667 tended to enrich the carbon metabolism processes at P1 and P2 and multiple stress response processes at P3 and P4. In D20170203, the DEGs mainly enriched carbon metabolism processes at the P1 stage and numerous stress-response pathways from P2 to P4. In particular, the DEGs significantly enriched photosynthesis-related pathways at P4 (in AKZ-NRGR667) and P3 (D20170203). In addition, the DEGs in both genotypes enriched the phenylpropanoid biosynthesis, plant signal transduction, carotenoid biosynthesis, diterpenoid biosynthesis, brassinosteroids biosynthesis, and nitrogen metabolism pathways at different stages of AKZ-NRGR667 and D20170203. Thus, these processes may be involved in growth and developmental processes regulating tillers.

### 2.5. Weighted Correlation Network Analysis Reveal Key Pathways at Specific Developmental Stages

A co-expression network of the DEGs with similar expression patterns revealed the stage-specific or phenotype-specific pathways between the high- and low-tillering orchardgrass at different developmental stages. The WGCNA generated eighteen distinct modules (labeled by different colors) (Figure 5a). Six modules (pink, magenta, tan, blue, green, and turquoise) correlated with specific phenotypes or stages according to the expression profiles of eigengenes (Figure 5b). For example, the pink module contained 410 highly expressed genes at the four-leaves stage (L4) of AKZ-NRGR667 (Figure 5e). The magenta (bud stage, L3 and L4) (Figure 5f), tan (L2, L3, and L4) (Figure 5g), and blue (L1, L2, and L3) modules (Figure 5c) had 353, 189, and 1662 highly expressed genes in high-tillering orchardgrass genotype, respectively. The green module encompassed genes predominantly expressed in the bud and L1 stages of D20170203 (Figure 5d). In particular, the turquoise module had 3165 highly expressed genes in all stages of D20170203 (Figure 5h).

KEGG revealed diverse functional categories in these six modules. In AKZ-NRGR667, genes in the pink module significantly enriched the nitrogen metabolism pathway (Appendix A). Most genes in the magenta module were highly associated with the plant-pathogen interaction, plant hormone signal transduction, and MAPK signaling pathways (Appendix A). In the tan module, the genes mainly enriched the butanoate metabolism and circadian rhythm pathways (Appendix A). Additionally, genes in the blue module significantly enriched photosynthesis-related and terpenoid backbone biosynthesis pathways (Appendix A). The DEGs in the green and turquoise modules were strongly associated with D20170203 and significantly enriched oxidative phosphorylation, amino and nucleotide sugar metabolism, and beta-alanine metabolism (Appendix A). These DEGs also enriched several SPL transcription factors, such as *DgSPL3*, *DgSPL6*, *DgSPL9*, and *DgSPL15*, that reportedly function in plant development [43]. These co-expression results showed that tiller development is highly correlated with the nitrogen metabolism, phytohormone signaling, photosynthesis, and carotenoid synthesis pathways in AKZ-NRGR667, and with the oxidative phosphorylation, amino- and ribose metabolism, and β-alanine metabolism pathways in D20170203.

The above results further confirmed several pathways, including photosynthesis, phytohormone signaling, carotenoid biosynthesis, diterpenoid biosynthesis, brassinosteroids biosynthesis, nitrogen metabolism, oxidative phosphorylation, and β-alanine metabolism between AKZ-NRGR667 and D20170203 at different developmental stages. Four important pathways significantly differed between the high- and low-tillering orchardgrass, including chlorophyll metabolism, lutein synthesis, nitrogen metabolism, and GA biosynthesis pathways.

### 2.6. Expression Pattern Identification of Key Tillering Regulators in Four Pathways

The chlorophyll metabolic pathway had eleven DEGs between AKZ-NRGR667 and D20170203 (Figure 6). The *HemE* encoding uroporphyrinogen III decarboxylase (UROD) was substantially expressed in AKZ-NRGR667 than D20170203 from the bud to the L4 stage. ALA dehydratase (ALAD), encoded by *HemB*, also showed a more substantial expression in AKZ-NRGR667 than in D20170203. However, differential expression of *HemB* occurred mainly from L1 to L4. Seven DEGs encoding enzymes, including GluRS, PBGD, MgCh, MgPMT, MgPEC, POR, and CS, were substantially expressed in AKZ-NRGR667 than in D20170203 from L1 to L3 stages.

*GluRS* was located upstream, while *POR* and *CHLG* were located downstream of the chlorophyll pathway, indicating the differences between AKZ-NRGR667 and D20170203 materials. The expression patterns of *HemA* (encoding GluTR) and *HemD* (encoding UROS) were almost opposite across the five developmental stages. *HemA* expression was higher in AKZ-NRGR667 than in D20170203, except at the L3 stage. In contrast, *HemD* expression was higher in D20170203 than in AKZ-NRGR667 across all stages. Thus, nearly all DEGs (except *HemA* and *HemD*) were more highly expressed in AKZ-NRGR667 than in D20170203, with the expression increasing from the bud to the L3 stage, then slightly decreasing at the L4 stage.

The qRT-PCR data were similar to the sequencing results (Appendix A), where almost all the nine key chlorophyll enzyme genes were highly expressed in AKZ-NRGR667 than in D20170203. This evidence suggests L3 is an important time point for tiller development. The inconsistency in tiller development may be caused by the higher efficiency of photosynthesis in AKZ-NRGR667 than in the D20170203.

Lutein, the most abundant carotenoid in higher plants, protects plants against photostress by stabilizing the photosystem assembly [44]. The first step of lutein synthesis involves precursor GGPP, synthesized as a phytoene through PSY catalysis. The lutein metabolism pathway involves *PSY*, *PDS*, and *ZDS* genes located upstream of the pathway (Figure 7). This study revealed five important enzyme-related genes of this pathway that were differentially expressed between the high- and low-tillering materials. In AKZ-NRGR667, *PSY* was up-regulated from L1, peaked at L2, and was down-regulated afterward.

In contrast, *PSY* was up-regulated from L3 and decreased at the L4 stage in AKZ-NRGR667 and D20170203. The expression of three differentially expressed downstream genes, *PD*, *ZDS*, and *LCYE* was highly similar to *PSY*. Cytochrome P450-type monooxygenase 97A, encoded by *CYP97A*, showed a higher expression in AKZ-NRGR667 than D20170203 from the bud to the L4 stage. Overall, lutein biosynthesis genes were highly expressed at most developmental stages (L1 to L3) of AKZ-NRGR667. However, the lutein and zeaxanthin content contradicted the transcript levels of these DEGs (Figure 7). The difference in lutein content between these two orchardgrass genotypes was likely due to specific physiological processes or synthesis of other metabolites.

Furthermore, we identified six DEGs involved in nitrogen metabolism. Nitrogen (N) is an essential element for proper growth and development of plants [45]. In AKZ-NRGR667, the DG3C00652-encoded nitrilase gene (*NIT*) was highly expressed at all five stages relative to D20170203. The expression of the five genes was significantly different between AKZ-NRGR667 and D20170203, and the other DEGs were extremely different between the two genotypes. For example, the nitrate/nitrite transporter gene (*NRT*) showed peak expression at stage L4 with an increasing tendency from the bud stage of AKZ-NRGR667. However, *NRT* was up-regulated much earlier and was higher in D20170203 after peaking at stage L4. Nitrate reductase (*NR*) expression was more specific than the other DEGs associated with nitrogen metabolism. In AKZ-NRGR667, *NR* expression was higher at the bud and L4 stages than in D20170203. The reverse was observed at the L3 stage, where *NR* expression was lower in AKZ-NRGR667. The dramatic expression changes in the short term may be due to the different developmental states in the high- and low-tillering orchardgrass. However, *NR* probably has a stronger temporal expression specificity between high- and low-tillering orchardgrass, eventually leading to phenotypic differences in their tillering capacity.

Another gene, ferredoxin-nitrite reductase (*NiR*), showed only one expression peak in AKZ-NRGR667 and D20170203 and was gradually up-regulated on both genotypes from the bud stage. *NiR* peaked at the L3 and L4 stages in AKZ-NRGR667 and D20170203, respectively. Two DEGs, glutamine synthetase gene (*GS*) and glutamate synthase gene (*GOGAT*), showed relatively high expression in the L4 stage of AKZ-NRGR667. The expression of *NRT*, *NR*, *NiR*, and *GS* at the stage L3 stage was higher in D20170203 than in AKZ-NRGR667, and the expression of *NR*, *GS*, and *GOGAT* was higher at L4 stage of AKZ-NRGR667 than in D20170203. Low nitrogen metabolism efficiency probably caused more glutamate and glutamine accumulation in D20170203. This evidence indicates a significant difference in the nitrogen metabolism and utilization of AKZ-NRGR667 and D20170203 (Figure 8).

Finally, we analyzed the pathways related to gibberellin synthesis and metabolism, and identified three important DEGs: *ent-copalyl diphosphate synthase* (*CPS*), *gibberellin 20 oxidase* (*GA20ox*), and *gibberellin 2beta-dioxygenase* (*GA2ox*). Enzyme CPS is key in the gibberellin biosynthetic pathway, catalyzing *ent*-Copalyl diphosphate (*en*t-CDP) synthesis from geranylgeranyl pyrophosphate (GGPP). This study showed lower *CPS* expression in AKZ-NRGR667 than in D20170203 from the bud to the L4 stage. In contrast, the expression of *GA20ox* (located downstream in this pathway), and *GA2ox* was higher in AKZ-NRGR667 relative to D20170203 from the bud to the L4 stage. These results show that high- and low-tillering orchardgrass accumulate different amounts of GA types at the early stages of bud growth and development (Figure 2 and Figure 8).

## 3. Discussion

### 3.1. Transcriptional Differences in Pigment-Related Genes Cause Different Photosynthetic Assimilation Capacities between High- and Low-Tillering Orchardgrass

Plant growth and development is largely a process of energy conversion and utilization. Photosynthesis is among the most important processes that convert solar energy into chemical energy (carbohydrates) in certain organelles (such as chloroplasts) [46]. Carbohydrates synthesized during photosynthesis provide energy for cell differentiation/growth and new tissue formation, driving plant growth and development [47,48]. Thus, plant growth affects the efficiency of photosynthesis and the tillering ability of orchardgrass. Chloroplasts, where photosynthesis takes place, contain large amounts of chlorophyll [49], the dominant pigment in green leaves. Therefore, pathways related to chlorophyll synthesis and metabolism regulate the photosynthetic rate, affecting plant growth indirectly [50]. Chlorophyll metabolism involves highly coordinated synergistic enzyme-catalyzed reactions coordinated by known genes [51]. Photosynthesis involves four main distinct sections, including the “common pathway and siroheme branch”, “Mg branch and heme branch”, “chlorophyll cycle”, and “chlorophyll degradation pathway” [52]. Briefly, glutamyl-tRNA synthetase (GluRS) catalyzes initial substrate glutamate by forming glutamyl-tRNAGlu in the presence of ATP [53]. The glutamyl-tRNAGlu synthesizes 5-aminolevulinic acid (ALA), the first common precursor for protoporphyrin synthesis. Then, glutamyl-tRNA reductase (GluTR) and glutamate-1-semialdehyde aminotransferase (GSA-AT) catalyze ALA synthesis from glutamyl-tRNAGlu via a two-step reaction mediated by intermediate Glutamate 1-semialdehyde (GSA) [54].

Glutamyl-tRNA reductase is the first rate-limiting enzyme during ALA biosynthesis and adjusts ALA content [55]. In this study, the expression of *gltX*-encoded GluRS and *HEMA1*-encoded GluTR was significantly different between AKZ-NRGR667 and D20170203 [56,57], resulting in higher ALA levels in AKZ-NRGR667 than in D20170203. Loss-of-function GluTR mutants had reduced GluTR contents, conferring chlorophyll deficiency phenotypes and significantly affecting plant seeding development [58,59]. These results suggest that the two orchardgrass genotypes had different initial chlorophyll synthesis capacities or efficiencies, significantly affecting tiller formation and development.

Subsequently, the common chlorophyll and protoheme precursor, protoporphyrin IX, was synthesized from ALA through a series of enzymatic reactions. ALA dehydratase (encoded by *hemB*), porphobilinogen deaminase (encoded by *hemC*), uroporphyrinogen-III synthase (encoded by *hemD*), uroporphyrinogen III decarboxylase (encoded by *hemE*), coproporphyrinogen III oxidase (encoded by *hemF*), and protoporphyrinogen oxidase (encoded by *hemG*) catalyzed the enzymatic reactions [60]. ALA dehydratase (ALAD) functions in the early stages of chlorophyll biosynthesis [61]. In cotton, reducing ALAD activity enhances ALA accumulation in proportion to the degree of programmed cell death or plant development [62]. ALAD catalysis mediates porphobilinogen (PBG) synthesis, subsequently forming hydroxymethylbilane (HMB) from four molecular PBG catalyzed by the hemC-encoded porphobilinogen deaminase (PBGD). Subsequently, d-porphyrin ring isomerization form uroporphyrinogen III, which is catalyzed by uroporphyrinogen III decarboxylase (UROD) to form coproporphyrinogen III [63]. There is evidence that ALAD converts ALA to protoporphyrin IX in rate-limiting steps during chlorophyll synthesis, and the hemC-encoded PBGD may be a key regulator of this process [64]. Previous research showed that expressing essential biosynthesis genes (*hemA*, *hemL*, *hemB*, *hemC*, *hemD*, and *hemH*) and assembling rate-limiting enzymes (ALAD, PBGD, and UROD) at proper ratios efficiently enhances heme synthesis [65]. This study showed that higher expressions of genes encoding key enzymes increase chlorophyll in orchardgrass.

Chlorophyll synthase (ChlG) catalyzes a terminal reaction in the chlorophyll biosynthesis pathway, inducing Chl a formation and Chl b production through the chlorophyll cycle [66]. Overall, chlorophyll synthesis involves 15 or more enzymatic steps [67]. This study revealed that over two-thirds of these enzyme reaction steps, including key rate-limiting genes like *hemB*, *hemC*, and *GUS1*, significantly differ between the high-and low-tillering orchardgrass. The significant difference probably results from a cascade of accumulated amplification effects of individual genes in this pathway, which may occur before the plant morphology is established. Chlorophyll contents can largely represent the physiological status of plants [68], demonstrating a strong relationship to plant photosynthetic functions and photosynthetic capacities [69]. A lack of chlorophyll may reduce the absorption of light energy and inhibit plant growth by reducing the net photosynthetic rate [70]. These effects, including tillering, are particularly prominent throughout the plant growth and development stages. Low chlorophyll metabolism increases the production of new shoots and leaves, consuming more energy from photosynthesis and negatively affecting tillering. Current literature has focused on the effect of environmental factors on plant growth. However, studies on how the content and biosynthesis of chlorophyll directly affect plant differentiation are a few. Chlorophyll metabolism is regulated at various levels for adaptation to varying chlorophyll requirements during the growth and development of orchardgrass. This study showed that misregulating chlorophyll metabolism may reduce tiller capacity.

Besides chlorophyll, carotenoids are another predominant natural pigment responsible for the yellow, orange, and red colors of plants. Carotenoids absorb light at wavelengths beyond the efficient range for chlorophyll, such as blue-green to green wavelengths. In most green leaves, lutein comprises approximately half of the total carotenoids and dominates photosystem II [71,72]. Lutein accumulates in thylakoid membranes of chloroplasts and is mainly associated with light harvesting and photoprotection as they bind to specific proteins in the light-collecting complexes of chloroplasts [73,74]. Chlorophyll content and light intensity positively correlate with photosynthesis before reaching the saturation point. Moreover, increasing the light intensity beyond this saturation point produces ROS and inhibits photosynthesis.

Sufficient lutein and zeaxanthin can maintain normal photoprotection and catalyze thermal dissipation of excess excitation energy [75,76] because lutein effectively transforms the excess excitation energy to harmless heat to avoid photosynthetic damage [77]. The initial tillers are formed under the cover of leaf sheaths. Thus, maximum lutein contents may be generated earlier than chlorophyll in the absence of light. Some reports have suggested that the earlier development of photosystem II than photosystem I in leaf tissues maximizes lutein concentrations before other pigments [78]. Kale, dill, and lettuce alter the ratio of lutein to other pigments early (during leaf development) [78,79,80], indicating that pigments may be unevenly distributed both spatially and temporally during the early stages of orchardgrass photosynthetic morphogenesis. Moreover, orchardgrass is a typical cool-season perennial grass with a high shade tolerance, and 50% to even 80% shading does not significantly reduce its dry weight [81]. Shade orchardgrass species have high lutein contents, and the lutein to xanthophyll ratio is strongly correlated to the shade tolerance index [82]. Therefore, chlorophyll and lutein combinations can utilize different light wavelengths to produce energy, especially under shades or in the early stage of orchardgrass development when chlorophyll synthesis is inadequate.

### 3.2. The Nitrogen and GA Metabolism Synergistically Regulate Tiller Development

In plastids, geranylgeranyl diphosphate (GGPP) synthase (GGPPS) produces GGPP, a common precursor for chlorophyll, carotenoid, gibberellin biosynthesis, and other vital metabolites [83]. Phytoene synthase (PSY) catalyzes phytoene formation from GGPP, the first committed step of carotenogenesis [84]. Desaturation and isomerization reactions of 15-cis-phytoene desaturase (PD) and zeta-carotene desaturase (ZDS) produce lycopene from phytoene via intermediates ζ-carotene [85]. Lycopene cyclization produces α-carotene and β-carotene, which undergo subsequent hydroxylation and epoxidation to generate lutein and zeaxanthin, respectively. Lycopene epsilon-cyclase (Lcy ε) and lycopene beta-cyclase (Lcy β) are the two important rate-limiting enzymes in these steps [86].

Moreover, GGPP is a direct precursor of GA, and ent-copalyl diphosphate synthase (CPS) catalyzes a committed step in GA biosynthesis to produce ent-Copalyl-PP in plastids. Kaurene synthase (KS) catalyzes a cyclization reaction initiated by pyrophosphate ionization to produce kaurene [87]. Two soluble 2-oxoglutarate-dependent dioxygenases, GA 20-oxidase (GA20ox) and GA 2-oxidase (GA2ox), catalyze the last two steps of GA biosynthesis, directly affecting bioactive GA levels in plants [88]. The transcriptional data from this study showed that mRNA levels of all the above-mentioned genes significantly differed between the high- and low-tillering orchardgrass, probably explaining the differences in lutein, zeaxanthin, and GA contents in their leaves.

More importantly, the biosynthesis and accumulation of carotenoids and chlorophylls are coordinately increased during the photomorphogenesis of higher plants [89,90]. The light activates the functional photosynthetic apparatus, and the macroscopic manifestation of these cumulative intrinsic molecular differences could explain the major difference in tiller formation in the early stages of orchardgrass development. The regulation of plant growth and development by bioactive GAs is more evident than in chlorophyll and carotenoids, which indirectly regulate plant growth and development through photosynthesis. Excessive GA accumulation inhibits tiller initiation; thus, active GA synthesis and degradation determine the appropriate levels needed to maintain normal growth and development. In rice and wheat, overexpressing tall fescue *FaGA2ox4* exhibited GA-deficiency phenotypes like increased tiller and root number [91,92]. More importantly, synergistic GA-phytohormone regulation is crucial, especially with strigolactones. GA affects strigolactone biosynthesis, thus regulating tillering [93]. Chlorophyll, carotenoids, and GAs share a common precursor, GGPP. Therefore, the transcriptional differences in chlorophyll, carotenoid, and GAs pathways are far more pronounced than the effects of single pathway differences on orchardgrass tillering.

Besides carbon assimilation through photosynthesis, light incidence may directly regulate plant development and growth by affecting nitrogen metabolism indirectly [94]. Nitrogen is among the most essential inorganic nutrients for higher plants, the main component of chlorophyll and photosynthetic apparatus [95]. The nitrogen content in leaves positively affects photosynthesis, possibly due to the nitrogen partitioning in the photosynthetic enzymes, pigment content, and chloroplast formation [96,97]. Meanwhile, proper rations of red and blue light could increase the activities of enzymes involved in nitrogen metabolism, such as nitrite reductase [98]. Increasing the proportions of red light promotes nitrogen uptake and accelerates biomass accumulation [99,100].

In contrast, this study indicated significant differences in photosynthesis-related and nitrogen metabolism pathways at early stages of development, hinting that these pathways may be intrinsically linked to orchardgrass tillering. Nitrogen metabolism involves two main processes: extracellular nitrate/nitrite transmembrane transport and intracellular oxidation-reduction [101]. The nitrate transporter (NRT) is pivotal in nitrate/nitrite absorption, transportation, and assimilation [102]. In this study, the expression of NRT genes was higher in AKZ-NRGR667 than in D20170203, suggesting that nitrate/nitrite uptake may be more efficient in D20170203. However, almost all DEGs related to intracellular nitrogen metabolic pathways showed higher expression in AKZ-NRGR667 than in D20170203, contrary to the expression pattern of the NRT gene related to the extracellular transporter.

Therefore, the ability of orchardgrass to absorb extracellular nitrogen and intracellular nitrogen metabolism may vary across orchardgrass with different tillering capacities. Nitrate/nitrite may accumulate in large quantities, but the expression of genes related to enzyme activity and nitrogen metabolism may determine nitrogen utilization [103]. A consecutive two-step reaction by nitrate reductase (NR) and nitrite reductase (NiR) reduces nitrate into ammonium [101]. Finally, glutamine synthetase (GS) and glutamate synthetase (GOGAT) convert ammonium to glutamate, which participates in subsequent plant metabolism [104]. After the intracellular conversion of inorganic nitrogen into organic nitrogen, the synthesis of proteins implicated in cell wall/cytoskeleton synthesis, cell division, and cell growth stimulates plant growth [105]. In this process, inefficient intracellular conversion of nitrate/nitrite and nitrogen deficiency possibly have similar effects on plants. In Arabidopsis, the loss of *atnrt2.1-1* function strongly decreased the shoot biomass and shoot-to-root ratio, inhibiting shoot growth [106]. Limited nitrogen supply affects the shoot-root resource allocation in favor of root development [107]. In AKZ-NRGR667, genes encoding key enzymes involved in nitrogen metabolism maintained high expression levels, leading to more tiller formation. Nitrogen assimilation requires a lot of energy and carbon (C) sources; thus, efficient photosynthetic pigment synthesis and functioning of photosynthesis are necessary for the C/N balance during plant growth.

Initial studies revealed that nitrogen regulates tillering in a relatively independent process in cereals, such as rice and wheat [108,109]. Thus, few studies have analyzed the crosstalk between nitrogen metabolism and GA signaling [110]. Several regulators, including GRF4, NGR5, NPF, DELLA, GA2ox, and KS2, function as key nodes in the crosstalk [111,112,113,114]. GROWTH REGULATING FACTOR 4 (GRF4) is a positive regulator of various nitrogen-metabolism genes in rice. GRF4 accelerates nitrogen assimilation by regulating gene-encoding enzymes, such as GS and GOGAT. GRF4 physically interacts with DELLA proteins, growth inhibitors of the GA signaling pathway, thus integrating nitrogen metabolic regulation and GA-mediated plant growth [114]. A recent study demonstrated that GA and nitrogen metabolism regulate rice tillering through NITROGEN-MEDIATED TILLER GROWTH RESPONSE 5 (NGR5) and the APETALA2 (AP2) transcription factor [113]. DELLA proteins competitively inhibit GID1-NGR5 interaction, maintaining a higher *NGR5* abundance and ultimately enhancing tillering via improved nitrogen assimilation [113]. This evidence supports the pivotal role of GA signaling in GRF4- and NGR5-mediated regulation of tillering. However, GA3 and nitrogen may show opposite effects on plant tillering. The key issue to be explored in the future is how to balance the spatial and temporal distribution of GAs and nitrogen to improve nitrogen use efficiency. Overall, the results of this study were mainly based on biochemical and transcriptomic data, and thus lacked comprehensive evidence from multiple perspectives. Therefore, the key genes reported in this study should be confirmed and validated, both in vitro and in vivo, in future studies.

## 4. Materials and Methods

### 4.1. Plant Material and RNA Extraction

High-tillering orchardgrass genotype AKZ-NRGR667 and low-tillering orchardgrass genotype D20170203 were chosen for this study. RNA extraction, plant materials screening, and validation were conducted according to our previous study [36]. Forty-five single uniformly sized tillers of AKZ-NRGR667 and D20170203 with similar morphology were randomly divided into three groups (15 single tillers per group). The tillers were grown in silica sand supplemented with Hoagland nutrient solution in a greenhouse under a temperature regime of 22 °C/15 °C day/night, a photoperiod of 14 h/10 h day/night, and 70% relative humidity. The first sampling was conducted when buds had reached to 0.5 cm in height, defined as the bud stage (HB, bud stage of high-tillering orchardgrass AKZ-NRGR667; LB, bud stage of low-tillering orchardgrass genotype D20170203). The second sampling was performed when the first main leaf unfolded, defined as the one-leaf stage (L1). The third, fourth, and fifth samplings were performed when the second, third, and fourth main leaves unfolded, defined as two-leaf (L2), three-leaf (L3), and four-leaf (L4) stages, respectively. Accordingly, we defined the bud (HB and LB) to one-leaf (L1) stage as phase 1 (P1), one-leaf (L1) to two-leaves (L2) stage as phase 2 (P2), two-leaves (L2) to three-leaves (L3) stages as phase 3 (P3), and three-leaves (L3) to four-leaves stages (L4) as phase 4 (P4). “A” denoted the high-tillering orchardgrass (AKZ-NRGR667), and “D” represented the low-tillering orchardgrass (D20170203). Thus, A_P1 to A_P4 represented the four developmental stages of the high-tillering orchardgrass genotypes, and D_P1 to D_P4 represented the four developmental stages of the low-tillering orchardgrass genotype.

### 4.2. Transcriptome Sequencing and Data Analysis

Total RNA was extracted using the TRIzol reagent (Invitrogen, Carlsbad, CA, USA), following the manufacturer’s instructions. The RNA samples were treated with DNaseI to remove DNA, resolved in 1% agarose gel, and their purity was verified using the NanoPhotometer^®^ spectrophotometer (IMPLEN, Westlake Village, CA, USA). The RNA concentration was measured using the Qubit^®^ RNA Assay Kit in the Qubit^®^ 2.0 Fluorimeter (Life Technologies, Carlsbad, CA, USA). While, RNA integrity was assessed using the RNA Nano 6000 Assay Kit of the Agilent Bioanalyzer 2100 system (Agilent Technologies, Santa Clara, CA, USA). Afterward, all RNA samples with 260/280 ratios of 1.8–2.0, 260/230 ratios of 2.0–2.5, and an RNA integrity value > 8.0 were selected for cDNA library construction and sequencing.

Approximately 3 μg of RNA from each sample was used for cDNA library generation using the NEBNext^®^Ultra™ Directional RNA Library Prep Kit for Illumina^®^ (NEB, Ipswich, MA, USA), following the manufacturer’s recommendations. Thirty cDNA libraries were sequenced at the Novogene Bioinformatics Institute (Beijing, China) on Illumina’s Novaseq 6000 platform (Illumina, San Diego, CA, USA). Clean reads were mapped onto the orchardgrass reference genome using Hisat2 (v2.0.5) and were searched and annotated using BLASTx (v2.14.0) with a 1 × 10^−5^ E-value cut-off. The read count per gene was expressed as the expected number of FPKM base pairs sequenced. The FPKM values were plotted using a box plot to depict the global abundance of gene expression.

Differentially expressed genes (DEGs) were identified via pairwise sample comparisons in adjacent sampling points (A_L1 vs. A_HB, A_L2 vs. A_L1, A_L3 vs. A_L2, A_L4 vs. A_L3, D_L1 vs. D_LB, D_L2 vs. D_L1, D_L3 vs. D_L2, and D_L4 vs. D_L3) using the DESeq R package (1.18.0) [115].

Genes with an adjusted |log_2_ (fold change)| > 1 and *p*-value < 0.05 were assigned as differentially expressed. These DEGs were subjected to GO enrichment analyses using the GOseq R package, with *p*-values of <0.05 as the cut-off for the significantly enriched GO [116]. The KEGG database (http://www.genome.jp/kegg/, accessed on 1 January 2023) and KOBAS software (v2.0) were used to test the statistically enriched KEGG pathways, with *p*-values of <0.05 as the threshold value of the significantly enriched pathways [117].

The WGCNA package was used for the weighted gene co-expression network analysis in R (v3.3.0) using 25,071 genes with FPKM values of >0 from at least three sampling points [118]. A gene expression adjacency matrix was constructed for analyzing network topology at a soft thresholding power = 10. The blockwise module was used to obtain the modules at default settings. An eigen-gene network was constructed to represent the module relationships. Then, principal component analysis (PCA) and hierarchical clustering were performed to assess the similarities between the biological replicates. The PCA was performed using the expressed genes in different samples in the R program following default parameters. However, hierarchical clustering was performed via the complete linkage method, showing the gene expression changes across different stages [35]. A heat map was obtained using the OmicShare tools (www.omicshare.com/tools, accessed on 1 December 2016).

### 4.3. Verification by qRT-PCR

The RNA-seq data were validated by quantitative real-time PCR (qRT-PCR) using nine chlorophyll biosynthesis-related genes, including *GluRS* (*DG2C01560*), *GluTR* (*DG5C05347*), *HemB* (*DG7C03037*), *HemC* (*DG2C02319*), *MgCh* (*DG2C03690*), *MgPMT* (*DG3C06102*), *MgPEC* (*DG5C0263*), *POR* (*DG1C07069*), and *CHLG* (*DG6C02553*). First-strand cDNAs were synthesized using the MonScript™ RTIII All-in-One Mix with dsDNase (Monad, Wuhan, China). The qRT-PCR reaction was performed on the Bio-Rad CFX96 system (Bio-Rad, Hercules, CA, USA) using the MonAmp™ SYBR Green qPCR Mix (Monad, Wuhan, China), as instructed by the manufacturer. Each sample had three biological and three technical replicates, and glyceraldehyde *3-phosphate dehydrogenase* (*GAPDH*) was the reference gene. The relative expression was calculated using the 2^−ΔΔCT^ method as previously described for orchardgrass [35]. All primers are listed in Appendix A.

### 4.4. Photosynthetic Parameters and Chlorophyll Content

The chlorophyll (Chl) contents of the high- and low-tillering orchardgrass were measured as previously described [119]. The photochemical efficiency (Fv/Fm) and performance index on an absorption basis (PIABS) were determined using a Pocket PEA chlorophyll fluorescence system (Hansatech Instruments, Pentney, UK). The stomatal conductance (Gs), net photosynthetic rate (Pn), and transpiration rate (Tr) were measured using a portable CIRAS-3 photosynthesis system (PP Systems, Amesbury, MA, USA) that provided 400 μL L^−1^ CO_2_ and 800 μmol photon m^−2^ red and blue light in the leaf chamber. Finally, the water use efficiency (WUE) was calculated as Pn/Tr [120]. The chlorophyll content and photochemical parameters were measured at the four-leaf (L4) stage.

### 4.5. The Measurement of Lutein, Zeaxanthin, Glutamate, Glutamine, and GAs

The concentrations of lutein, zeaxanthin, glutamine, and GAs in orchardgrass were determined at the four-leaf (L4) stage using enzyme-linked immunosorbent assay (ELISA), with minor modification [121,122,123]. Briefly, solid phase antibodies for lutein, zeaxanthin, glutamine, and GAs were added to 96-well plates, and then 40 μL of the sample dilution buffer and 10 μL of the sample were added to each well. The plates were sealed and incubated at 37 °C in a water bath for 30 min. Thereafter, the sealing membrane was removed, the liquid discarded, and the plate was filled with the washing solution for thirty seconds and washed five times. The secondary antibodies labeled with horseradish peroxidase were added to each well, and the plates were incubated for 30 min at 37 °C. The plates were then washed, and the samples were incubated in 3,3′,5,5′-tetramethylbenzidine substrate (TMB) for 30 min at room temperature, after which the reaction was stopped by adding 2 N sulfuric acid solution. The absorbance of each sample was measured using the SpectraMax ABS plus Microplate Reader (Molecular Devices, San Jose, CA, USA) at 450 nm. The glutamate concentration was determined using the glutamic acid kit (Baiao Laibo, Beijing, China) following the manufacturer’s instructions.

## 5. Conclusions

In summary, we assessed photosynthetic parameters, partially essential intermediate biomolecular substances, and the comprehensive transcriptome to elaborate their profiles during the early stage of tiller development in high- and low-tillering orchardgrass genotypes. The photosynthetic efficiency and morphological development of high- and low-tillering orchardgrass differed significantly at the early stage of tiller formation. Stage-specific expression patterns provided clues that signal transduction and energy-related metabolism, especially photosynthetic-related processes, may dramatically influence tiller induction and development. WGCNA and functional enrichment identified distinctively co-expressed gene clusters and four main regulatory pathways, including chlorophyll, lutein, nitrogen, and GA metabolism pathways. Hence, photosynthesis, carbohydrate synthesis, efficient nitrogen utilization, and phytohormone signaling pathways may be closely and intrinsically linked at the transcriptional level (Figure 9). These findings enhance our understanding of tillering in perennial grasses, providing a new breeding strategy to improve biomass and nitrogen use efficiency in forages.

## Figures and Tables

**Figure 1 ijms-24-16370-f001:**
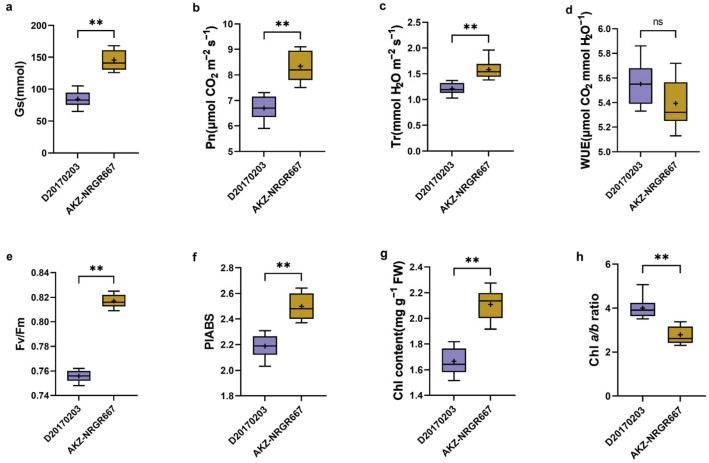
Photosynthetic parameters and chlorophyll content. (**a**) Stomatal conductance (Gs, mmol); (**b**) Net photosynthetic rate (Pn, μmol CO_2_ m^−2^ s^−1^); (**c**) Transpiration rate (Tr, mmol H_2_O m^−2^ s^−1^); (**d**) Water use efficiency (WUE, μmol CO_2_ mmol H_2_O^−1^); (**e**) Photochemical efficiency (Fv/Fm); (**f**) PIABS; (**g**) Chlorophyll content (Chl, mg g^−1^ FW); (**h**) Chlorophyll a/b ratio. “ns” indicates no significant difference, and “**” indicates the statistical significance at *p*-value < 0.01.

**Figure 2 ijms-24-16370-f002:**
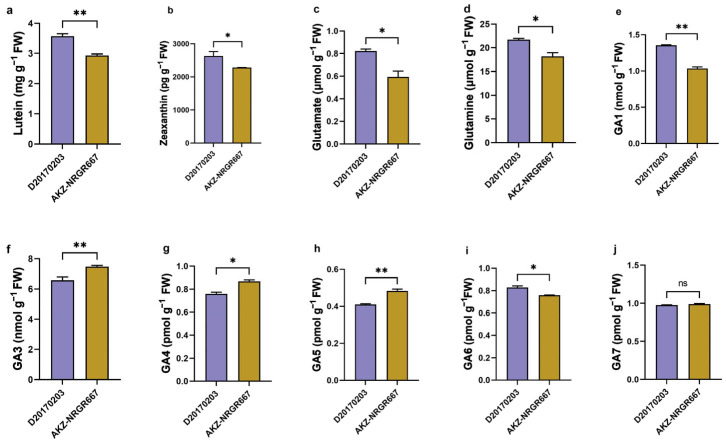
The concentrations of lutein, zeaxanthin, glutamate, glutamine, and gibberellic acid (GA) in the high- and low-tillering orchardgrass. (**a**) Lutein content (mg g^−1^ FW); (**b**) zeaxanthin content (mg g^−1^ FW); (**c**) glutamate content (μmol g^−1^ FW); (**d**) glutamine content (μmol g^−1^ FW); (**e**) GA1 content (nmol g^−1^ FW); (**f**) GA3 content (nmol g^−1^ FW); (**g**) GA4 content (pmol g^−1^ FW); (**h**) GA5 content (pmol g^−1^ FW); (**i**) GA6 content (pmol g^−1^ FW); (**j**) GA7 content (pmol g^−1^ FW). “ns” indicates no significant difference, “*” indicates the statistical significance at *p*-value < 0.05, and “**” indicates the statistical significance at *p*-value < 0.01.

**Figure 3 ijms-24-16370-f003:**
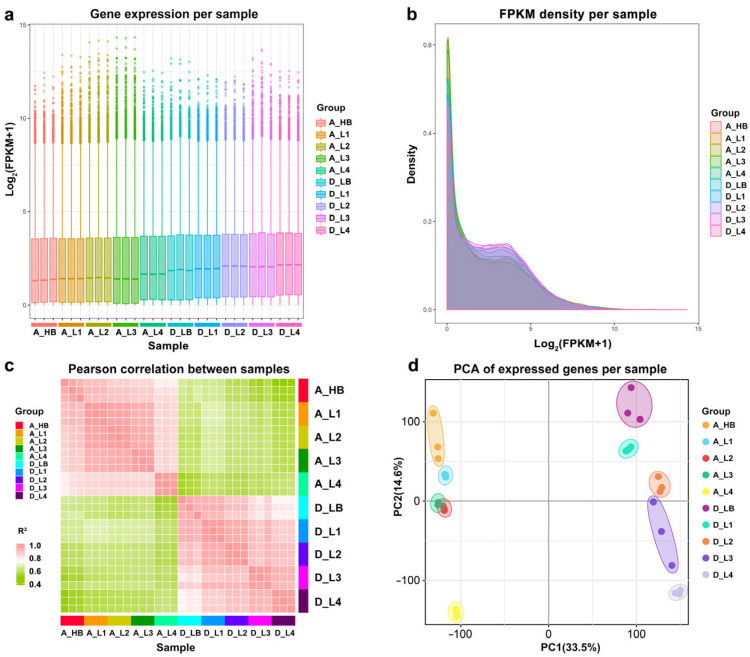
Transcriptional relationships between samples. (**a**) Gene expression per sample; (**b**) FPKM (fragments per kilobase of transcript per million mapped reads) density per sample; (**c**) Pearson correlation between samples; (**d**) principal component analysis of expressed genes per sample (showing ten distinct clusters). HB, bud stage of the high-tillering orchardgrass (AKZ-NRGR667); LB, bud stage of the low-tillering orchardgrass (D20170203); L1, one-leaf stage; L2, two-leaves stage; L3, three-leaves, and L4, four-leaves stage.

**Figure 4 ijms-24-16370-f004:**
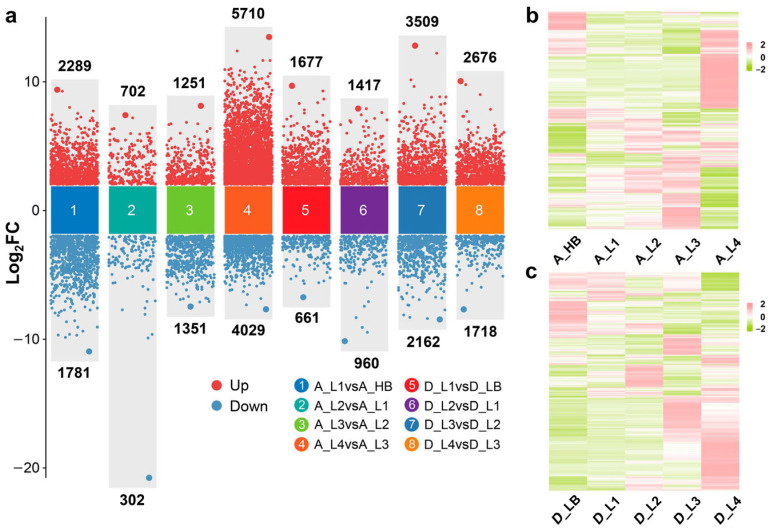
Differentially expressed genes (DEGs) between samples. (**a**) The number of up- and down-regulated genes in eight pairwise sampling stages, including A_L1 vs. A_HB, A_L2 vs. A_L1, A_L3 vs. A_L2, A_L4 vs. A_L3, D_L1 vs. D_LB, D_L2 vs. D_L1, D_L3 vs. D_L2, and D_L4 vs. D_L3; (**b**) the heatmap showing the DEGs from each ‘AKZ-NRGR667′ sample group based on the average FPKM (fragments per kilobase of transcript per million mapped reads) of biological replicates; (**c**) the heatmap showing the DEGs from each ‘D20170203′ sample group based on the average FPKM of biological replicates.

**Figure 5 ijms-24-16370-f005:**
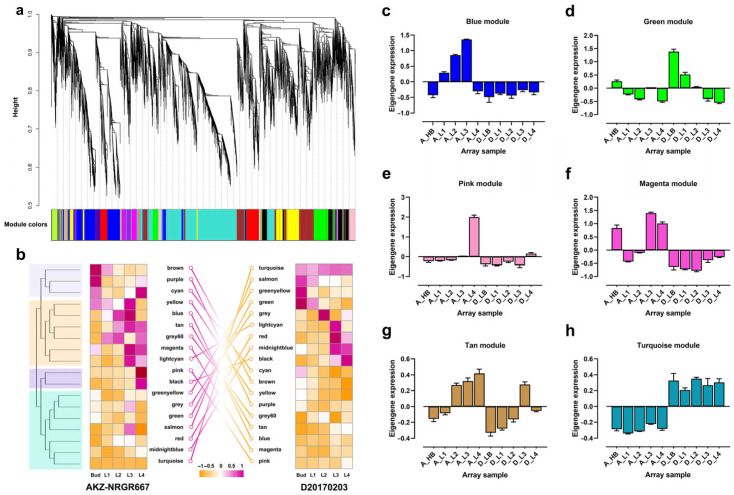
A weighted correlation network analysis (WGCNA) of genes. (**a**) A hierarchical cluster tree showing the co-expression modules identified via WGCNA. Each leaf in the tree represents one gene. Major tree branches constitute 18 modules labeled by different colors. (**b**) The relationship between modules and samples. The heatmap shows the correlation between different modules. The deeper the red color, the higher the correlation. The colors on the left indicate the clusters in AKZ-NRGR667 after the correlation analysis between samples and modules. The connecting lines indicate module correlation between AKZ-NRGR667 and D20170203 samples. (**c**–**h**) are the expression patterns of eigengene in six modules, including pink, magenta, tan, blue, green, and turquoise, respectively. The sample labels are the same as above.

**Figure 6 ijms-24-16370-f006:**
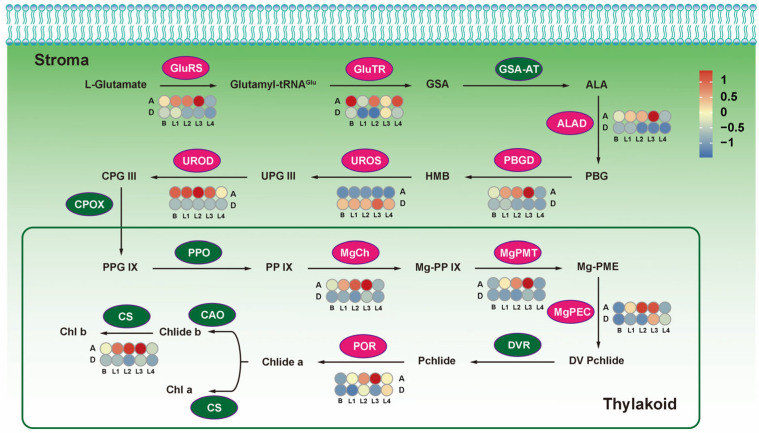
Chlorophyll metabolism in AKZ-NRGR667 and D20170203. The ellipses denote enzymes in different chlorophyll synthesis steps. Light magenta indicates that the genes were differentially expressed between the two genotypes in at least one stage, and green indicates no significant difference. Blue to red indicates the expression of genes that encode enzymes catalyzing corresponding biochemical reactions in different tissues. Sample labels are the same as mentioned above. ALA, 5-aminolevulinic acid; ALAD, ALA dehydratase; CAO, chlide a oxygenase; Chl a, chlorophyll a; Chl b, chlorophyll b; Chlide a, chlorophyllide a; Chlide b, chlorophyllide b; CPG III, Coproporphyrinogen III; CPOX, coproporphyrinogen III oxidase; CS, chlorophyll synthase; DV Pchlide, divinyl protochlorophyllide a; DVR, 3,8-divinyl protochlorophyllide an 8-vinyl reductase; GluRS, glutamate-tRNA ligase; GluTR, glutamyl-tRNA reductase; GSA, glutamate 1-semialdehyde; GSA-AT, glutamate-1-semialdehyde 2,1-aminomutase; HMB, hydroxymethylbilane; MgCh, magnesium chelatase subunit H; MgPEC, magnesium-protoporphyrin IX monomethyl ester cyclase; Mg-PME, magnesium protoporphyrin monomethyl ester; MgPMT, magnesium protoporphyrin IX methyltransferase; Mg-PP IX, magnesium protoporphyrin IX; PBG, porphobilinogen; PBGD, porphobilinogen deaminase; Pchlide, protochlorophyllide; PP IX, protoporphyrin IX; PPG IX, protoporphyrinogen IX; PPO, protoporphyrinogen oxidase; POR, protochlorophyllide reductase; UPG III, uroporphyrinogen III; UROD, uroporphyrinogen III decarboxylase; UROS, uroporphyrinogen-III synthase.

**Figure 7 ijms-24-16370-f007:**
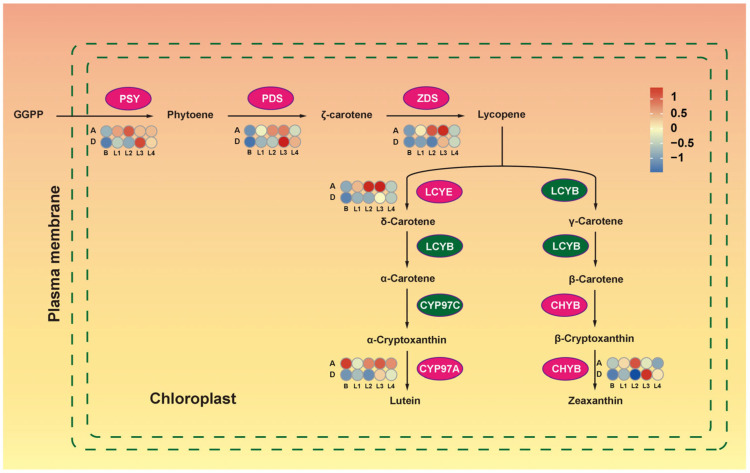
Lutein metabolism in high- and low-tillering orchardgrass. The ellipses denote enzymes in different steps of lutein metabolism. Light magenta indicates the differential expressed enzyme-encoding genes between the two genotypes in at least one stage. Green indicates no significant difference between the enzyme-encoding genes. Blue to red represents genes that encode enzymes catalyzing corresponding biochemical reactions in different tissues. Sample labels are the same as mentioned above. CHYB: beta-carotene hydroxylase; CYP97A, cytochrome P450-type monooxygenase 97A; CYP97C, cytochrome P450-type monooxygenase 97C; GGPP, geranylgeranyl diphosphate; LCYB, lycopene beta-cyclase; LCYE, lycopene epsilon-cyclase; PDS, 15-cis-phytoene desaturase; PSY, 15-cis-phytoene synthase; ZDS, ζ-carotene desaturase.

**Figure 8 ijms-24-16370-f008:**
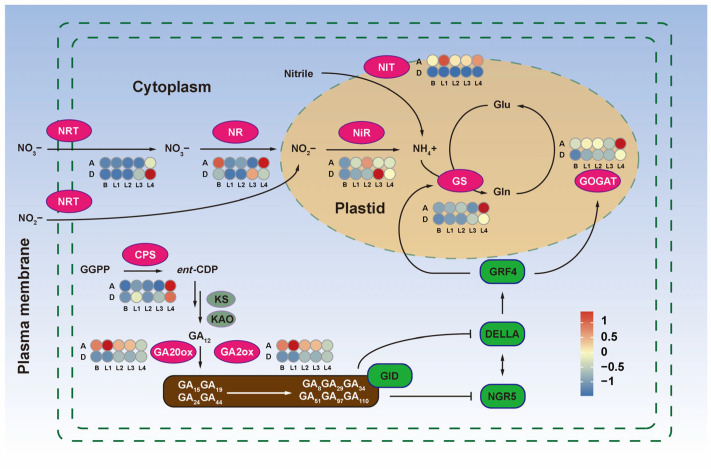
Nitrogen and gibberellic acid (GA) metabolism in high- and low-tillering orchardgrass. The ellipses denote enzymes at different steps of nitrogen and GA metabolism. Light magenta indicates differentially expressed enzyme-encoding genes between AKZ-NRGR667 and D20170203 in at least one stage. Grayish green indicates no significant difference in enzyme-encoding genes. Deep green denotes proteins related to nitrogen and the GA pathway. Sample labels are the same as mentioned above. Lines with arrows indicate positive regulation, and those with blunt ends indicate negative regulation. CPS, ent-copalyl diphosphate synthase; ent-CDP, ent-copalyl diphosphate; GA2ox, gibberellin 2beta-dioxygenase; GA20ox, gibberellin 20 oxidase; GGPP, geranylgeranyl pyrophosphate; GID, gibberellin insensitive dwarf; GOGAT, glutamate synthase; GRF4, growth-regulating factor 4; GS, glutamine synthetase; KAO, ent-kaurene oxidase; KS, ent-kaurene synthase; NGR5, nitrogen-mediated tiller growth response 5; NiR, ferredoxin-nitrite reductase; NIT, nitrilase; NR, nitrate reductase; NRT, nitrate/nitrite transporter.

**Figure 9 ijms-24-16370-f009:**
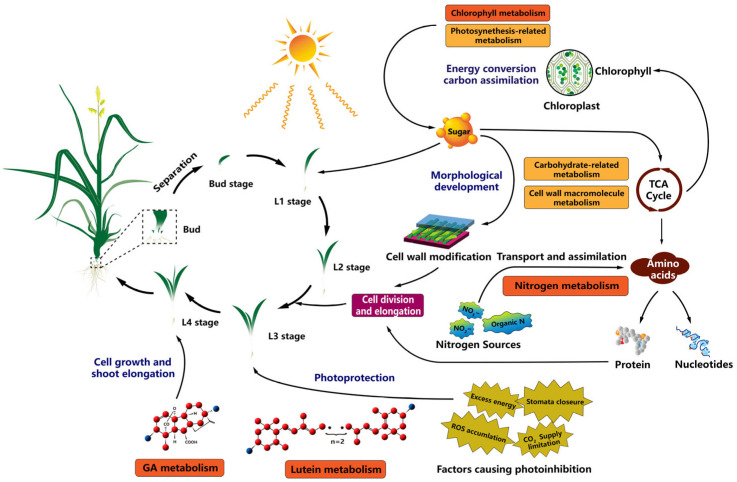
Schematic representation of the pathways involved in the early stage of tiller formation in orchardgrass. Chlorophyll, lutein, nitrogen, and gibberellic acid (GA) metabolism-induced energy conversion, carbon/nitrogen assimilation, morphological development, cell division/growth, and shoot elongation in the early tillering stage. Sample labels are the same as mentioned above.

## Data Availability

The raw sequence data have been deposited in the Genome Sequence Archive (Genomics, Proteomics & Bioinformatics 2021) at the National Genomics Data Center (Nucleic Acids Research 2022), China National Center for Bioinformation/Beijing Institute of Genomics, Chinese Academy of Sciences (GSA: CRA011917) and are publicly accessible (https://ngdc.cncb.ac.cn/gsa, accessed on 20 July 2023).

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
