# Peer review of "Transcriptome Profiling Provides Insights into the Early Development of Tiller Buds in High- and Low-Tillering Orchardgrass Genotypes"

_ijms, 2023, doi:10.3390/ijms242216370_

Round 1
Reviewer 1 Report
Comments and Suggestions for Authors
Manuscript with the title “Transcriptome Profiling Provides Insights into the Early Development of Tiller Buds in High- and Low-tillering Orchardgrass Genotypes” assessed photosynthesis, quantified some essential biomolecules and profiled the transcriptome of high/low-tillering orchardgrass at early stage of tiller development. Some results confirmed some previous general findings and others clarified some aspects in regards with perennial forage grasses. The results are promising and have potential practical implications in breeding high-yield genotypes.
General notes
The full name with author Dactylis glomerata L., shall be given only at the beginning of the manuscript, afterwards as D. glomerata.
The novel contribution or the authors following the results of their study must be more clearly delimited (particularly in conclusions and also in the abstract). From what I see it particularly revolves around the comparative nature of the study between high/low-tillering genotypes.
Line 62 - species name shall be with lowercase letters
Line 127. D. glomerata with italics
Best regards.
some syntax and minor grammar improvements
Author Response
Thanks for your review.
- We have modified “Dactylis glomerata L” as “D. glomerata” after its first appearance.
- We have revised all the species name in this manuscript with lowercase letters, and “D. glomerata” with italics also revised throughout the manuscript.
- As you concerned the syntax and minor grammar, we have edited the language of manuscript by native English speakers to ensure better readability.We have checked all the abbreviations in this manuscript. Meanwhile, we also edited the language of manuscript by native English speakers to ensure better readability.

Reviewer 2 Report
Comments and Suggestions for Authors
The Authors present a research regarding the transcriptome profiling in different genotypes of Orchardgrass (Dactylis glomerata L.) to study the early development of tiller buds.
This plant is very important both from an environmental and an economic point of view as it represents fodder for farm animals. Consequently, the implications of the knowledge of the metabolism and physiology of this plant are numerous and useful. The study is very in-depth and the data presented in the paper are many, interesting and thoroughly discussed, in relation to a large and sufficiently updated bibliography. The figures are explanatory and the supplementary materials are also helpful.
My suggestion is to make the introduction more usable/accessible even to those who are less familiar with these topics, checking that all abbreviations are explained at the first citation, perhaps adding a table in which names are reported and functions of the various genes/proteins mentioned .
It is a good work, illustrative for other studies of this kind
Minor editing of English language is required
Author Response
Thanks for your review. We have checked all the abbreviations in this manuscript. Meanwhile, we also edited the language of manuscript by native English speakers to ensure better readability.

Reviewer 3 Report
Comments and Suggestions for Authors
Here are my comments for the manuscript:
- The introduction provides a good overview of tillering in grasses and prior work in model species like rice, but could better highlight the knowledge gaps specifically for perennial grasses like orchardgrass.
- The objectives, while appropriate, could be more clearly stated early in the introduction.
- The materials and methods are clearly described, but some additional details would be helpful:
- Specify the growth conditions (light, temperature, etc) for the plant material more precisely.
- Provide information on sequencing depth and genome coverage for the RNA-seq data.
- Explain criteria for selecting differentially expressed genes (DEGs) and enriched pathways.
- The results are extensive but could be better organized to improve flow and readability. Consider restructuring some sections.
- The discussion provides good interpretation of the results, but lacks critical analysis of the limitations of the approaches used. The authors should comment on any caveats of the transcriptomic analysis.
- The proposed model figure highlights the major findings, but a more integrated model showing connections between all the pathways and regulators would strengthen the conclusions.
- The writing is clear in most parts, but grammatical/language editing is needed in some areas. There are missing words, awkward phrasings, and typos to address.
- The figures are appropriate, but many axis labels, legends, and text are too small or blurry to read easily.
- The manuscript makes a useful contribution to understanding tillering in perennial grasses, but revisions to improve clarity, thoroughness, and polish are needed prior to publication.
Comments on the Quality of English Language- The writing is clear in most parts, but grammatical/language editing is needed in some areas. There are missing words, awkward phrasings, and typos to address.
Author Response
Thanks for your review.
1. The introduction provides a good overview of tillering in grasses and prior work in model species like rice, but could better highlight the knowledge gaps specifically for perennial grasses like orchardgrass.
Response: We have added the description about the knowledge gaps after the introduction of our work on tillering of orchardgrass line 101 to l03 as below “Although these studies provide clues to tillering regulation in orchradgrass, their main conclusions still make reference to the tillering mechanisms in annual model crops.” Meanwhile we also added the description of tillering study on orchardgrass in line 97 to line 102.
2. The objectives, while appropriate, could be more clearly stated early in the introduction.
Response: We have stated our objectives “Consequently, revealing the transcriptional difference between high- and low-tillering orchardgrass genotypes is one of the major concerns of this study.” in second paragraph of early introduction part line 53 to 54, page 2.
- The materials and methods are clearly described, but some additional details would be helpful:
- Specify the growth conditions (light, temperature, etc) for the plant material more precisely.
Response: we have modified the description of growth condition as “The tillers were grown in silica sand with Hoagland nutrient solution in a greenhouse at temperature regime of 22 ℃/15 ℃ day/night, a photoperiod of 14 h/10 h day/night and 70% relative humidity.” line 568 to 570, page 18.
- Provide information on sequencing depth and genome coverage for the RNA-seq data.
Response: we have added the details of sequencing depth for the RNA-seq data as “A total of 206.41 Gb of data were generated, with an average of 6.88 Gb data per sample (Supplementary material 2)” line 146 to 147. And genome coverage for the RNA-seq was describe as “Over 88.24% of the clean reads were mapped to the reference genome, and 84.98% were mapped to unique genomic locations (Supplementary material 3). Moreover, 87.86, 3.89, and 8.25% of the clean reads mapped to exons, introns, and intergenic regions, respectively (Supplementary material 4)” line 153 to 156.
- Explain criteria for selecting differentially expressed genes (DEGs) and enriched pathways.
Response: we have added criteria for selecting differentially expressed genes (DEGs) and enriched pathways as “Genes with an adjusted |log2 (fold change) | > 1 and p-value < 0.05 were assigned as differ-entially expressed. The significant DEGs were subjected to GO enrichment analyses in the GOseq R package, with p-values < 0.05 as the cut-off of significantly enriched GO [116]. The KEGG database (http://www.genome.jp/kegg/) and KOBAS software were used to test the statistically en-riched KEGG pathways, with p-values < 0.05 as the threshold value of significantly enriched pathway [117].” line 603 to line 608.
- The results are extensive but could be better organized to improve flow and readability. Consider restructuring some sections.
Response: we have rearranged our result in our manuscript, including move “Photosynthetic parameters and chlorophyll content” and “Lutein, zeaxanthin, glutamate, glutamine, and GAs concentration” to the beginning of the results section; replaced the original Figure 5 with the Supplementary Figure S6; renumbered the figures in this manuscript.
- The discussion provides good interpretation of the results, but lacks critical analysis of the limitations of the approaches used. The authors should comment on any caveats of the transcriptomic analysis.
Response: we have added the relevant descriptions in our modification as below “Overall, this study was mainly based on biochemical results and transcriptomic data, comprehensive evidence from multiple perspectives is insufficient. Meanwhile, the confirmation of key candidates is absent, and validation, both in vitro and in vivo, needs to be addressed in future studies.” line 558 to line 561.
- The proposed model figure highlights the major findings, but a more integrated model showing connections between all the pathways and regulators would strengthen the conclusions.
Response: Thanks very much for your suggestions. Initially, we planned to construct an integrated model to show the connections between all the pathways and regulators. However, we have not identified the function of too many genes involved in the pathways, and there is no articles related it. Therefore, we constructed a simplified model to display the key regulatory pathways at different stages of early tillering, as showed in Figure 9.
- The writing is clear in most parts, but grammatical/language editing is needed in some areas. There are missing words, awkward phrasings, and typos to address.
Response: We have edited the language of manuscript by native English speakers.
- The figures are appropriate, but many axis labels, legends, and text are too small or blurry to read easily.
Response: we have re-edited the axis labels, legends, and text in figures.
- The manuscript makes a useful contribution to understanding tillering in perennial grasses, but revisions to improve clarity, thoroughness, and polish are needed prior to publication.
Response: We have restructured, edited the language, and improved the quality of the images of manuscripts.
- Comments on the Quality of English Language: The writing is clear in most parts, but grammatical/language editing is needed in some areas. There are missing words, awkward phrasings, and typos to address.
Response: We have re-edited the language of this manuscripts.
